# Role of Exopolygalacturonase-Related Genes in Potato-*Verticillium dahliae* Interaction

**DOI:** 10.3390/pathogens10060642

**Published:** 2021-05-23

**Authors:** Xiaohan Zhu, Mohammad Sayari, Fouad Daayf

**Affiliations:** 1Department of Plant Science, University of Manitoba, 222 Agriculture Building, Winnipeg, MB R3T2N2, Canada; huhu772@hotmail.com (X.Z.); Mohammad.Sayari@umanitoba.ca (M.S.); 2Agriculture and Agri-Food Canada Saskatoon Research Centre, Saskatoon, SK S7N 0X2, Canada

**Keywords:** polygalacturonase, pectinase, *Verticillium dahliae*, virulence factor, pathogenicity factor

## Abstract

*Verticillium dahliae* is a hemibiotrophic pathogen responsible for great losses in dicot crop production. An *ExoPG* gene (VDAG_03463,) identified using subtractive hybridization/cDNA-AFLP, showed higher expression levels in highly aggressive than in weakly aggressive *V. dahliae* isolates. We used a vector-free split-marker recombination method with PEG-mediated protoplast to delete the *ExoPG* gene in *V. dahliae.* This is the first instance of using this method for *V. dahliae* transformation. Only two PCR steps and one transformation step were required, markedly reducing the necessary time for gene deletion. Six mutants were identified. *ExoPG* expressed more in the highly aggressive than in the weakly aggressive isolate in response to potato leaf and stem extracts. Its expression increased in both isolates during infection, with higher levels in the highly aggressive isolate at early infection stages. The disruption of *ExoPG* did not influence virulence, nor did it affect total exopolygalacturonase activity in *V. dahliae*. Full genome analysis showed 8 more genes related to polygalacturonase/pectinase activity in *V. dahliae*. Transcripts of *PGA* increased in the *△exopg* mutant in response to potato leaf extracts, compared to the wild type. The expression pattern of those eight genes showed similar trends in the *△exopg* mutant and in the weakly aggressive isolate in response to potato extracts, but without the increase of *PGA* in the weakly aggressive isolate to leaf extracts. This indicated that the *△exopg* mutant of *V. dahliae* compensated by the suppression of ExoPG by activating other related gene.

## 1. Introduction

*Verticillium dahliae* is a hemibiotrophic pathogen that causes wilt symptoms and results in great losses in a wide range of dicot hosts [1], including in important economic crops [2]. Traditional control management strategies, such as crop rotation and the use of green manure, are not very effective in preventing Verticillium wilt [3,4]. Soil fumigation results in effective Verticillium wilt reduction, but has a negative impact on the environment [5,6,7]. Genetic resistance has not been very useful so far, i.e., although tomato lines contain the *Ve* resistance gene, it only has resistance to *V. dahliae* race 1 [8,9], and cultivars with effective resistance have not been developed in the majority of the crops at risk. 

The plant cell wall is a key barrier of protection from pathogen attacks. It consists primarily of cuticular wax, cutin, glycans, cellulose, pectic substances, and cell wall structural proteins [10]. However, many microbes produce various cell wall-degrading enzymes (CWDEs) to degrade these plant cell-wall components. CWDEs include cutinases, pectinases, cellulases, glucanases and proteases [10]. Glycosidic bonds of polysaccharides can be cut by enzymes belonging to the glycoside-hydrolases (GH) family [11]. Polygalacturonases belong to these pectin-degrading enzymes, which include both endopolygalacturonases (EndoPGs) and exopolygalacturonases (ExoPGs). EndoPGs hydrolyze the polysaccharide randomly to produce oligogalacturonides, while ExoPGs hydrolyze the polymer from the non-reducing end to produce a single galacturonic acid [12,13]. Both the EndoPGs and ExoPGs belong to the GH28 family [12]. 

Cell walls of dicot plants contain a large amount of pectin [11,14]. In parallel, pathogens of dicots contain more pectin-degrading enzymes-encoding genes than those infecting monocots [11]. However, the function of PGs in pathogenicity seems to be more specific to certain types of diseases, rather than to the class of pathogens [11]. *Alternaria citri* and *A. alternata* “rough lemon pathotype” are two pathogens for citrus with similar morphology [15]. *A. citri* causes Alternaria black rot, while *A. alternata* causes Alternaria brown spot [15]. The Endo-PGs from the two pathogens show a high similarity in nucleotide identity (99.6%) and biochemical properties [15]. Disruption of the *EndoPG* gene in *A. citri* significantly decreased the magnitude of Alternaria black rot symptoms in citrus and maceration symptoms in potato tissue, while disruption of *Endo-PG* gene in *A. alternata* did not cause any change in virulence, and indicates that pectin-degrading enzymes may play different functions and are involved in various processes [15]. 

Using a subtractive hybridization cDNA-AFLP method, El-Bebany et al. [16] found that a *ExoPG* gene (VDAG_03463) was up-regulated in the highly aggressive *V. dahliae* isolate (Vd1396-9) while it was down-regulated in a weakly aggressive isolate (Vs06-14), in response to root extracts from both moderately resistant and susceptible potato cultivars. This infers that this *ExoPG* gene (VDAG_03463) may play a role in the pathogenicity of *V. dahliae*. 

A common method to investigate fungal gene functions is targeted gene-disruption. In *V. dahliae*, the most popular method for disrupting a target gene is the vector-based *Agrobacterium*-Mediated T-DNA insertion method [17,18]. However, a vector-free split-marker recombination technique, developed for gene deletion in *Saccharomyces cerevisiae* and applied to *Magnaporthe oryzae,* also seems attractive [19,20]. Catlett et al. [20] prepared two DNA fragments, each containing a DNA flanking of the targeted gene with half of the selected marker gene. These were introduced into fungal protoplast cells to obtain the knockout mutant by homologous recombination between the overlapping regions of the half marker gene region in the two DNA fragments and the flanking region of the target gene [20]. PEG-mediated protoplast transformation is a common gene transfer method for many other fungi and plants [21,22,23]. Even though *V. dahliae* protoplast transformation may be time-saving and has been used in a few studies [24,25], it is not as popular as *Agrobacterium*-mediated transformation. 

In this study, we used a vector-free split-marker recombination method with an optimized PEG-mediated protoplast transformation technique to knock out *VdExoPG* gene in *V. dahliae*. Our objectives were to: (1) determine the expression of *ExoPG* gene under induction with potato tissues and during *in-planta* infection; (2) introduce the vector-free split-marker recombination method to obtain *ExoPG* mutants in *V. dahliae*; (3) determine the pathogenicity of the mutants; (4) determine the total exopolygalacturonase activity of both the mutants and the wildtype; and in case *ExoPG* gene’s mutation does not affect pathogenicity, (5) compare the expression levels of other polygalacturonase/pectinase-encoding genes in *ExoPG* mutants, wild type as well as weakly aggressive isolate in response to potato stem or leaf extracts to explain the polygalacturonase/pectinase activity compensation in the mutants. 

## 2. Results

### 2.1. ExoPG Expression against Potato Extracts

The *ExoPG* gene’s response was measured under treatment with extracts from potato leaves, stems and roots. As shown in Figure 1, *ExoPG* responded more in the highly aggressive isolate Vd1396-9 to potato leaf and stem extracts, compared to in the weakly aggressive isolate Vs06-07. However, gene expression in both highly and weakly aggressive isolates were up-regulated in response to root extracts with no significant difference between the two. 

The expression of the *ExoPG* gene during infection of potato detached leaves was significantly higher in the highly aggressive isolate Vd1396-9 at 3 DAI, when compared to the weakly aggressive one (Figure 1). 

### 2.2. Mutant Production

Following the procedure showed in Figure 2, the 715 bp from the upstream region and the 713 bp from the downstream region of the selected *ExoPG,* as well as the 789 bp from the 5′ segment and 909 bp from the 3′ segment of Hygromycin resistant gene (*Hph*) gene were both successful amplified by PCR (Figure 3). Both 5′ terminator of primers FS-AR and FS-BF were flanked with a 27 bp nucleotide sequence from 5′ and 3′ terminator of the *Hph* gene sequence, respectively. The PCR product of the upstream region of the selected *ExoPG* region and the 5′ segment of the *Hph* gene were used as templates to use the overlapping PCR (Figure 2; Table 1). During the first eight cycles of the overlapping PCR, these two DNA fragments used each other’s overlapping regions as a “primer” to elongate a new DNA fragment. The primer (Primer: FS-AF/ HY-R; Table 1) can attach to both ends of the new fragment, to enrich the new DNA fragment (Figure 2; Table 1). The same procedure was performed with the PCR product of the downstream region of the selected *ExoPG* region and 3′ segment of *Hph* gene with primer YG-F/ FS-BR (Figure 2; Table 1 and Table 2). As shown in Figure 3, the 1486 bp from DNA fusion of the upstream region of the selected *ExoPG* region and the 5′ segment of *Hph* gene, the 1622 bp from DNA fusion of the downstream region of the selected *ExoPG* region and the 3′ segment of the *Hph* gene, were both successfully amplified. After transforming both fused DNA from overlapping PCR into the protoplast by PEG-mediated transformation (Figure 2), seven positive knock-out transformants were obtained from 40 transformants identified by PCR (Figure 4). In addition, six single hygromycin gene replacement mutants of the seven transformants were identified by Southern blot (Figure 4). The mutants *△exopg-ko-18* and *△exopg-ko-23* were randomly chosen for subsequent experiments. 

### 2.3. Characterization of △exopg Mutant

#### 2.3.1. Growth Rate, Conidiation and Microsclerotia Formation 

As shown in Figure 5, there were no significant differences in the growth, conidiation, or formation of microsclerotia between the *△exopg* mutants *(**△exopg-ko-18*, *△exopg-ko-23*) and wildtype Vd1396-9. 

#### 2.3.2. Pathogenicity 

To determine the difference in aggressiveness between the *△exopg* mutants and wildtype (Figure 6), all isolates were inoculated onto the susceptible potato cultivar Kennebec. There was no dramatic change in total area under disease progress curve (Total AUDPC) for infection or disease severity, or plant height and vascular discoloration measurements. 

#### 2.3.3. Total Exopolygalacturonases Activity of *△exopg* Mutant

To determine the influence of *ExoPG*’s disruption in total exopolygalacturonase activity, the supernatant from cultured isolates *(**△exopg-ko-18,*
*△exopg-ko-23*, weakly aggressive isolate Vs06-07 and wildtype Vd1396-9) were collected over time and assayed for enzyme activity. Pectin was added to the CDB medium to stimulate the production of ExoPG. As shown in Figure 7, the total exopolygalacturonase activity in all tested isolates under the elicitation of pectin was significantly higher than the control group, however, there was no significant differences between the *△exopg* mutants and wildtype at almost all time points. On the contrary, the total exopolygalacturonase activity of weakly aggressive isolate Vs06-07 after treatment with pectin was significantly lower than wildtype Vd1396-9 (Appendix A). 

### 2.4. Identification of Other Polygalacturonase/Pecninase Coding Genes and Expression Pattern of These Genes

Polygalacturonase/pectinase-related genes included in the genome of *V. dahliae* were designated (i.e., Polygalacturonase A (PGA) to PGD and Pectinase 1 (PEC1) to PEC4). Since disruption of the *ExoPG* coding gene did not affect the virulence of *V. dahliae*, the expression levels of these genes (Accessions #: VDAG_07608, VDAG_02879, VDAG_08097, VDAG_05992, VDAG_09366, VDAG_08098, VDAG_01781, and VDAG_00768) were determined in wild type, weakly aggressive isolate (Vs06-07) and the mutant *△exopg*, with elicitation of Kennebec potato leaves or stems extracts. Most of the genes in water control treatment were down-regulated in the mutant *△exopg* compared to the wild type (Figure 8). When responding to both or one of the potato extracts, the expression level of most of the genes, except *PGA* and *PEC1*, showed significant a decrease in both isolates, compared to that in wild type with water control treatment (Figure 8). However, the expressions of *PGA* showed a drastic increase in the mutant *△exopg* compared with the wild type, in response to potato leaf extracts (Figure 8). The expression of *PGA* in response to stem extracts, and *PEC1* to leaf extracts, both showed more significant increases in the wild type than in the mutant △exopg (Figure 8). Even given that most genes expression in both wild type and mutant were down-regulated in response to potato extracts, the expression level of *PGC*, *PGD*, *PEC2* and *PEC3* show a higher level in the mutant *△exopg* than the wild type in response to one or both potato extracts (Figure 8). Interestingly, *PGA* and *PGD* showed a significantly lower level in the mutant *△exopg* than in the wild type under elicitation of stem extracts, but the opposite occurred after elicitation with leaf extracts (Figure 8). Our results indicate that different other polygalacturonase/pectinase genes may compensate that of *ExoPG* transcripts under different conditions, and among them, *PGA* might play a significant role for compensation (Figure 8).

The expression of *PGC* and *PEC3* showed a more significant increase in the weakly aggressive isolate than the highly aggressive wild type stain when exposed to stem extracts (Figure 9). The number of transcripts of *PGB*, *PGC*, *PGD* and *PEC3* in weakly aggressive strain were significantly higher than highly aggressive wild type when exposed to one or both extracts (Figure 9). On the other hand, the expression levels of *PGA* in response to stem extracts and *PEC1* to leaf extracts both increased drastically in the highly aggressive wild type compared to in the weakly aggressive isolate (Figure 9). 

When comparing the expression patter of all eight polygalacturonase/pectinase genes, between the highly aggressive wild type and the mutant *△exopg*, as well as between the highly aggressive wild type and weakly aggressive isolate, in most cases, similar trends were seen in the mutant *△exopg* and weakly aggressive isolate in response to potato extracts (Figure 8 and Figure 9). However, the *PGA* is interesting among all these genes, as it showed an obvious increase in the mutant *△exopg* exposed to the leaf extracts, but did not significantly change in weakly aggressive isolate, when both are compared to highly aggressive wild type (Figure 8 and Figure 9). The expressions of *PGC* and *PEC3* in weakly aggressive isolate were significantly increased, but this was not the case in the mutant *△exopg* (Figure 8 and Figure 9). 

## 3. Discussion

The most common transformation method in *V. dahliae* is *Agrobacterium*-mediated transformation, in both disruption and overexpression of a targeted gene [18,29,30,31,32,33]. However, PEG-mediated protoplast transformation is a useful method in other fungi and plant transformation [19,21,22,23]. Here, we successfully used this method in *V. dahliae*. Using this method, we successfully obtained 40 trans-DNA transformants, including ectopic strains and targeted gene mutation strains. 

Gene disruption in fungi, by either *Agrobacterium*-mediated transformation or PEG-mediated protoplast transformation, often requires vector construction [18,23,34,35]. In particular, *Agrobacterium*-mediated transformation requires a special vector in the procedure [33,36,37]. Constructing a vector is time-consuming and requires many steps. Here, we employed a vector-free split-marker recombination method for knocking out target genes in *V. dahliae*, as previously carried out in *S. cerevisiae* [20] and *M. oryzae* [19]. This method can be easily processed by only two steps of PCR, and therefore can typically be performed in one day. This method can substantially reduce the time and cost when compared to at least one-week requirement for constructing a vector for a targeted gene. Combining both vector-free split-marker recombination methods with PEG-mediated protoplast transformation reduces the total time for manipulation, including transformation, to as little as three days. This is drastically shorter than the time needed for vector-based *Agrobacterium*-mediated transformation. In addition to time consideration, this method could reduce the number of transformants that contain randomly inserted exotic DNA in the fungal genome. If each of the two fusion DNA fragments (flanking DNA with half marker gene) were separately inserted into the fungal genome, the transformants cannot grow on the selection media, because of the non-intact marker gene. Only when the two fusion DNA fragments are inserted in the proper place can the homologous recombination occur between overlapping regions of the two split-markers. The overlapping PCR in this study is a convenient and quick method to fuse different DNA fragments together, without finding or adding appropriate restriction enzyme sites in the edge of different DNA fragments. 

Pathogens can overcome plant cell wall protection by producing cell wall-degrading enzymes (CWDEs) [10]. In *A. citri*, an *EndoPG* gene is required for pathogenicity [15]. El-Bebany et al. [16] found that when *V. dahliae* responds to root extracts from both moderately resistant and susceptible potato cultivars, the expression of *ExoPG* (VDAG_03463) was up-regulated in the highly aggressive *V. dahliae* isolate (Vd1396-9), but down-regulated in a weakly aggressive one (Vs06-14) [16]. Based on these results, we wanted to investigate if the function of this *ExoPG* was involved in pathogenicity or interaction with the host. The degree of up or down-regulation in *V. dahliae* isolates with differential aggressiveness levels were not provided by El-Bebany et al. [16], since the experiment was conducted using a subtractive hybridization cDNA-AFLP method. 

In this study, we determined the fold-change of *ExoPG* expression in differentially aggressive *V. dahliae* isolates in response to potato leaf, stem, or root extracts, and during plant tissue infection. *ExoPG* responded more in the highly aggressive isolate Vd1396-9 to potato leaf and stem extracts, compared to in the weakly aggressive one. *ExoPG* was up-regulated in both isolates elicited with root extracts. However, *ExoPG* was up-regulated more in the highly aggressive isolate than in the weakly aggressive one at early infection stages. This may indicate that this gene may be activated by potato extracts in both isolates, but at a higher level in the highly aggressive one, and therefore may not play a primary role in infection. Subsequent comparison of the *△exopg* mutants and wildtype proved that *ExoPG* is not indispensable for *V. dahliae*’s pathogenicity. The total exopolygalacturonase activity in *V. dahliae* did not change due to the disruption of the *ExoPG* gene. However, when compared to the weakly aggressive strain vd06-07 of *V. dahliae*, wildtype isolate showed much higher *ExoPG* activity. So, it seems that other genes or pathways rather than the knocked out gene might help with the pathogenicity of highly aggressive isolate of *V. dahliae*. 

Comparison of the expression patterns of the other polygalacturonases/pectinases indicates that most of those genes showed no difference or decreased in the mutant *△exopg,* compared to wild type without elicitation, but *PGA* showed a more significant increase in the mutant *△exopg* than the wild type under elicitation with potato leaf extracts. Even though the expression of most of the genes in response to potato extracts showed down-regulation in both mutant and wild type, most genes still showed relatively higher levels in the mutant compared to the wild type. Similar trends in most genes were observed between weakly and highly aggressive isolates, except for *PGA* which does not show the same increase as that in mutant in response to leaf extracts. This indicates that these polygalacturonases/pectinases coding genes, *PGA* in particular, may compensate for the function of the knocked out gene in the *△exopg* mutant during interaction with potato, and this particular *ExoPG* is not a crucial virulence factor in *V. dahliae*. The increased activity of *PGA* in the mutant could also explain the reason why disruption of *ExoPG* had no effect on the virulence of transformants. Interestingly, although most of the polygalacturonases/pectinases showed a similar response to potato leaf and stem extracts in the mutant *△exopg*, activity of *PGA* showed an opposite trend with potato leaf or stem extracts in the mutant, indicating that those polygalacturonases/pectinases may also have various interactions in response to infection of different host tissues. In *Fusarium oxysporum*, three polygalacuronase *PG1*, *PG5*, *PGX4* were identified, and none of these had an effect on virulence [38,39,40]. In *Aspergillus niger*, deletion of an exopolygalacturonase gene, *PGXB*, caused partial reduction in virulence on apple fruits, and six other polygalacturonase genes showed a higher level of expression in *ΔpgxB* mutant than in wild type [41]. In *Ralstonia solanacearum*, disruption of polygalacturonases, endoPG *PehA* or ExoPG *PehB*, could both reduce virulence on eggplant and tomato [42]. Both PehA and PehB function quantitatively, and play important roles in rapid colonization in the host’s vascular system [42]. In rice pathogen *Burkholderia glumae*, knocking out of polygalacturonase genes PehA or PehB could not affect the virulence on rice [43]. These studies showed species-dependent results by polygalacturonases. It seems that some polygalacturonases may play the most important roles in pathogenicity, while others do not. *V. dahliae* possesses many polygalacturonases, but obviously, the *ExoPG* tested in this study is not important for virulence, as other polygalacturonases, especially *PGA,* could compensate for its function. It will be interesting to investigate the functions of the *PGA* in *V. dahliae*, because it showed critically higher expression in *△exopg* than the wild type under elicitation with potato leaf extracts, and this did not exhibit in weakly aggressive isolate. One or more of these polygalacturonases may play a more central role for virulence under other conditions.

## 4. Materials and Methods

### 4.1. Fungi Isolates and Plant Material

Highly aggressive Vd1396-9 and weakly aggressive Vs06-07 *V. dahliae* isolates, and susceptible potato cultivar Kennebec were used in this study [44,45]. Kennebec seedlings (germinated for 10 days) were planted in a soil, sand, and peat moss mixture with a ratio of 12:4:1, with day/night temperatures of 22/18 °C and 16/8 h photoperiod in growth cabinet. 

### 4.2. ExoPG Expression in Response to Potato Tissue Extracts and on Inoculated Detached Leaves

The expression of *ExoPG* in the differentially aggressive *V. dahliae* isolates Vd1396-9 and Vs06-07 in response to elicitation with potato leaf, stem, and root extracts were performed as described by Zhu et al. [18]. Briefly, the potato leaf, stem, or root extracts were added into 7-day cultured *V. dahliae* isolates Vd1396-9 and Vs06-07, respectively, and after an additional one-week culturing, the fungal material was collected for RNA extraction. 

The expression of *ExoPG* in the same differentially aggressive *V. dahliae* isolates inoculated onto potato detached leaves was also assessed as described by Zhu et al. [18]. In brief, *V. dahliae* isolates Vd1396-9 and Vs06-07 were cultured on potato dextrose agar (PDA) for 3 weeks prior to harvesting the conidia by water flooding. Four-week-old detached leaves were inoculated by immersing their petioles into 1 mL of conidial suspensions of Vd1396-9 or Vs06-07, and sterilized water was used as a control. Four to six individual detached leaves from different plants were combined as one sample, and three samples were collected for each treatment at each selected time-point (3, 12, and 17 days after inoculation (DAI)). RNA extraction and Quantitative Real-Time RT-PCR were used here with primers (ExoPG-QRT-F/R; Table 2) following the protocol from Omega Fungal RNA kit (Omega Bio-Tek, Inc., Norcross, GA, USA) and SsoFast EvaGreen Super mix (Bio-Rad Lab, Philadelphia, PA, USA). The cycle parameters were as follows: initial denaturation at 95 °C for 30 s, followed by 40 cycles including denaturation at 95 °C for 5 s, then heating up to 60 °C for 30 s (annealing/extension); after that, a melting curve was generated by heating from 65 °C to 95 °C with increases of 0.5 °C and 5 s dwell time, as well as a plate read at each temperature.

### 4.3. Mutants Production

#### 4.3.1. Protoplast Preparation and Transformation

Preparation of the protoplast cells of *V. dahliae* (highly aggressive isolate Vd1396-9) was conducted following the description of Dobinson [25] and Yelton et al. [46], with some improvements. In brief, *V. dahliae* was cultured in PDB broth for 4 days at 24 °C without shaking, and the mycelium collected and finely ground using sterile mortar and pestle under a laminar flow hood. The ground mycelium was then re-cultured in fresh PDB broth containing 0.001% thiamine for an additional 14 hours in the shaker at 120 rpm. Mycelia were collected by filtration on miracloth and washed with mycelia buffer (10 mM NaPO_4_ pH 7.5; 10 mM EDTA pH 8.0; 1 mM dithiothreitol) 2–3 times. The mycelium was then re-suspended in 30 mL mycelia buffer and shaken for 2 h at 24 °C at 60 rpm. The culture was then centrifuged at 1900 g for 10 minutes to collect the mycelium, decanted the supernatant, then incubated in 15 mL OM buffer (1.2 M MgSO_4_, 10 mM NaPO_4_ pH 5.8) with 10 mg/mL lysing enzymes from *Trichoderma harzianum* (Sigma-Aldrich Canada Co., Oakville, ON, Canada) and shaken overnight at 30 °C at 65 rpm. The suspension was finally filtered by miracloth and transferred into a 50 mL centrifuge tube, overlayed with 10 mL ST buffer (0.6 M sorbitol; 100 mM Tric-HCl pH 7.0), then centrifuged at 4 °C at 4000× *g* for 20 min. A glass Pasteur pipet was used to retrieve the 5–10 mL of protoplast cells in the interface between the OM and ST buffers. Two to 4 volumes of STC buffer (1.2 M sorbitol; 10 mM Tric-HCl pH 7.5; 10 mM CaCl_2_) were added, centrifuged at 4 °C, 4000× *g* for 20 min. The precipitate was washed 2 times with STC buffer, and finally re-suspended in 0.5 mL STC buffer with 8% PEG (PEG3350) or 8% DMSO, yielding a final protoplast concentration ranging from 3 to 5 × 10^7^ protoplast/mL.

We used the transformation protocol of Dobinson [25], with some modifications. An amount of 2 µg of DNA was added to 200 µL protoplast cell and incubated on ice for 20 min, followed by gentle addition of 0.625 mL PTC buffer (40% PEG 3350 in TSTC buffer (1 M sucrose; 50 mM Tris.HCl pH 8.0; 50 mM CaCl_2_)). The solution was mixed well following buffer addition, then incubated at room temperature for 20 min, followed by the addition of 5 mL complete media (CM) [47], containing 1 M sucrose, and shaking at 90 rpm, overnight at 24 °C.

An amount of 25 mL of CM medium (containing 1 M sucrose, 1% agar, 25 µg/mL Hygromycin B, 200 µg/mL Ampicillin) was added and gently mixed, then poured in a 15 cm-diameter petri dish. After solidifying, the media was overlaid with 50 mL CM medium (containing 1 M sucrose,1% agar, 75 µg/mL Hygromycin B, 200 µg/mL Ampicillin), incubated at 24°C for 7–14 days until the transformants grew to the surface. The transformants were then separated and transferred into fresh CM medium (1.5% agar, 25 µg/mL Hygromycin B, 200 µg/mL Ampicillin).

#### 4.3.2. Vector-Free Split-Marker Recombination Method for Knocking Out of ExoPG in *V. dahliae*

Vector-free split-marker recombination method requiring only two PCR steps followed by one transformation step was used in the present study [19,20]. Our aim was to knock out the front 1319 bp fragment of the *ExoPG* open reading frame (ORF) (1836 bp), but due to the proximity of the 3′ DNA sequence of *ExoPG* to another gene, we designed the downstream homologous recombination region so as not to interfere with the ORF of the other gene, but to cover a short part of that region with the *ExoPG* ORF. Briefly, as shown in Figure 2 and 3, the first step was to amplify the flanked upstream and downstream regions of the selected gene from the *V. dahliae* genome with the primers FS-AF/R and FS-BF/R (Table 2), then amplify the 5′ and 3′ segments of the *Hph* from pSK846 vector [30] with the primers Hph-F/HY-R, YG-F/Hph-R (Table 2), respectively. Both primer FS-AR and FS-BF were fused with a short sequence of 5′ terminator and 3′ terminator of the *Hph* gene (Figure 2). The amplified PCR product of the upstream region of the selected gene from *V. dahliae* and the 5′ segment of the *Hph* gene was used as a template for overlapping PCR (Table 1), in order to obtain the DNA fragment of the upstream region fused with the 5′ segment of *Hph*. Note the annealing temperature for the first 8 cycles of “self-primer” should be low in order to promote this combination. The downstream region of the selected gene from *V. dahliae* and the 3′ segment of the *Hph* gene was used as a template for overlapping PCR to obtain the DNA fragment of the downstream region fused with the 3′ segment of *Hph* (Figure 2). Finally, both overlapping PCR products of the upstream region fused with 5′ segment of *Hph* and the downstream region fused with the 3′ segment of *Hph* were transformed into protoplast cells of *V. dahliae* wild type Vd1396-9 (Figure 2). Only the homolog recombination which arose from both upstream and downstream regions of the selected gene, as well as the overlapping 5′ segment and 3′ segment of the *Hph* gene could produce positive transformants (Figure 2). All the transformants were selected on PDA plates with hygromycin B following the description of Zhu et al. [18]. The knocked out transformants were screened by PCR with primers ExoPG-ORF-F/R, ExoPG-UAF/Hph-TR (Table 2). Positive transformants were confirmed by Southern blot following Zhu et al. [18]. 

### 4.4. Mutants Characterization

#### 4.4.1. Pathogenicity Analysis of *△exopg* Mutant

The virulence of *△exopg* mutants were tested on potato cv. Kennebec following the description of Zhu et al. [18]. Briefly, Kennebec plants were grown in LA4 soil mix (SunGro Horticulture, Agawam, MA 01001, USA). After 21 days, plants were up-rooted, roots were washed and 1 cm was trimmed from the root tips, then they were inoculated with 10^6^ conidia/mL suspension of *△exopg* mutants *(**△exopg-ko-18,*
*△exopg-ko-23*), wild type, Vd1396-9 and ectopic control, *ExoPG-Ect-13* (fragment randomly inserted in *V. dahliae* genome without replacing the original *ExoPG* ORF) for 60 Sec. Plants were transplanted into 6-inch pots containing a mixture of pasteurized sand, soil and peat moss (16:4:1) and placed back into the controlled growth area for 2 weeks. Total area under the disease progress curve (total AUDPC) of percentage infection and disease severity was calculated weekly according to Zhu et al. [18]. Plant height measurements and vascular discoloration ratings were conducted at 5 weeks post-infection according to Alkher et al. [44].

#### 4.4.2. Growth Rate and Conidiation of *△exopg* Mutants

*△exopg* mutants (*△exopg-ko-18,*
*△exopg-ko-23*), wild type Vd1396-9 and ectopic control *ExoPG-Ect-13* were grown on PDA. The colony diameter was measured at 7 days and 16 days to calculate the growth of colony per day. Growth rate per day = (16 days colony diameter-7days diameter)/9 days. After 4 weeks, colony morphology photos were taken and conidia production was assessed according to Zhu et al. [18].

#### 4.4.3. Total Exopolygalacturonase Activity of *△exopg* Mutants

Total exopolygalacturonase activity of the *△exopg* mutants was measured according to Teixeira et al. [48], with some modifications. Briefly, *△exopg* mutants *△exopg*-ko-18, *△exopg-ko-23*, the weakly aggressive isolate Vs06-14, and the highly aggressive isolate Vd1396-9 control were cultured in Czapek-Dox Broth (CDB) media (Difco Laboratories, Sparks, MD, USA) with the addition of 1% pectin and shaking at 24 °C, 100 rpm. The same isolates cultured in CDB liquid media without pectin were used as controls. All treatments had 3 replicates. Samples of culture fluid (2 ml) were taken at 0, 2, 3, 4, 6, 10, and 14 days after inoculation (DAI) and filtered with miracloth. Samples were prepared by taking 50 μl from each culture filtrate and mixed with 450 μl reaction mixture (1% pectin in 25 mM sodium acetate buffer, pH 4.5). The mixture was then incubated at 45 °C for 10 minutes, then added to 500 μl DNS solution (dinitrosalicylic acid 10 g/L (1%), phenol 2 g /L (0.2%), sodium sulfite 0.5 g/L (0.05%), and sodium hydroxide 10 g/L (1%) [49], following boiling for 5 minutes. Total exopolygalacturonase activity was determined by recording the absorbance at 575 nm, which is measured by the release of galacturonic acid from pectin [50]. 

### 4.5. Identification of Other V. dahliae Polygalacturonase Related Genes and Their Expression Pattern 

We hypothesized that the suppression of *ExoPG* may be compensated by the activity of other polygalacturonase/pectinase coding genes. To test this hypothesis, using tBLASTn searches from the CLC genomics workbench (CLCBio, Aarhus Denmark), we could find 8 contigs possibly with polygalacturonase/pectinase genes within the available full genome sequence of *V. dahliae* [51]. The open reading frames (ORFs) encoded on these contigs (containing putative polygalacturonase/pectinase) were then put in Web AUGUSTUS (http://bioinf.uni-greifswald.de/augustus/, accessed on 1 May 2021) [52] for prediction. For further confirmation, identified ORFs were checked for signature domains of the target polygalacturonase/pectinase genes using InterProScan [53].

To determine the expression profile of the genes involved in polygalacturonase activity, samples were taken from seven-day-old cultures of Vd1396-9 (wildtype), Vd06-07 (weakly aggressive isolate) and *△exopg* under elicitation of Kennebec potato leaves or stems or water (control) extracts following Zhu et al. [18]. RNA was extracted from the freeze-dried samples and analyzed by quantitative RT-PCR (qRT-PCR) using CFX96 Thermal Cycler (BioRad, Hercules, CA, USA). The expression pattern of 8 different genes with polygalacturonase/pectinase activity (including *PGA*: VDAG_07608, *PGB*: VDAG_02879, *PGC*: VDAG_08097, *PGD*: VDAG_05992, *PEC1*: VDAG_09366, *PEC2*: VDAG_08098, *PEC3*: VDAG_01781 and *PEC4*: VDAG_00768) related to polygalacturonase activity was evaluated, using specific primer sets (Table 3) and Syber green qRT-PCR Master Mix kit (Bio-Rad Lab, Irvine, CA, USA) according to the manufacturer protocol. The cycle parameters were as follows: initial denaturation at 96 °C for 4 min, followed by 40 cycles including denaturation at 96 °C for 5 s, followed by heating up to 60 °C for 30 s (annealing/extension). In all applications, the *Histone H3* and Actin genes were used as the housekeeping genes (Table 2). All PCR reactions were carried out in triplicate. The analysis of data was performed using the 2^−ΔΔCt^ method [54].

### 4.6. Statistical Analysis 

PROC MIXED program processed by SAS Statistical Analysis Software (SAS Institute, Cary, NC, USA; release 9.1 for Windows) was used for data analysis in the current study. Some sets of data were applied with Log^10^ transformation for statistical analysis when necessary. The PROC UNIVARIATE was used to test the normality, all data qualified for normal distribution with Shapiro–Wilk test >0.9. The test for homogeneity was determined by residual comparison with studentized residuals’ critical values [55]. Mean values of all data were separated by least squared means and classified by the macro PDMIX800.sas [56] with α = 0.05 into a bunched letters result. Significant differences between different treatments were shown with totally different letters (*p* < 0.05).

## Figures and Tables

**Figure 1 pathogens-10-00642-f001:**
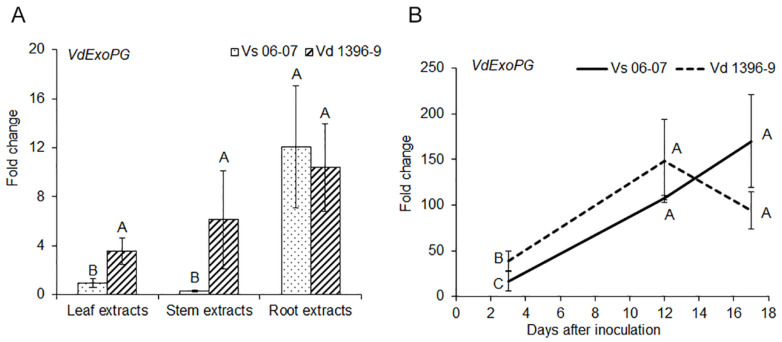
Expression of *ExoPG* gene in response to potato extracts and during infection. (**A**) Expression of *ExoPG* gene in response to potato extracts; (**B**) Expression of *ExoPG* gene during infection of detached Kennebec potato leaves. All qRT-PCR data were normalized with *V. dahliae* Histone H3. The bars (**A**) and point values (**B**) represent mean values (n = 3) ± standard error. Mean values marked by the same letters are not significantly different according to multiple comparison (Fisher’s LSD) test (*p* < 0.05).

**Figure 2 pathogens-10-00642-f002:**
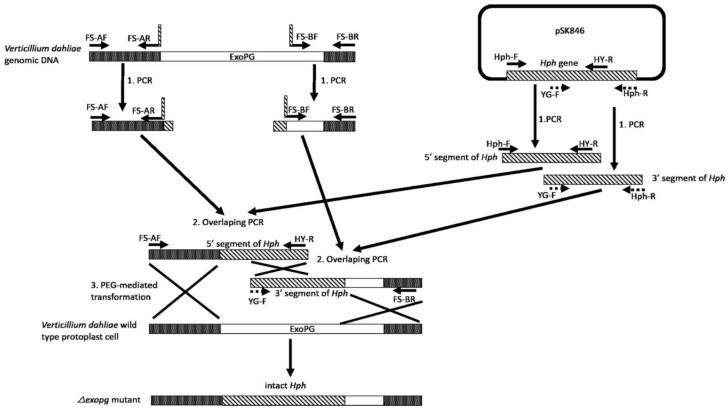
Procedure for vector-free-based knocking out of *V. dahliae*’s *ExoPG* gene.

**Figure 3 pathogens-10-00642-f003:**
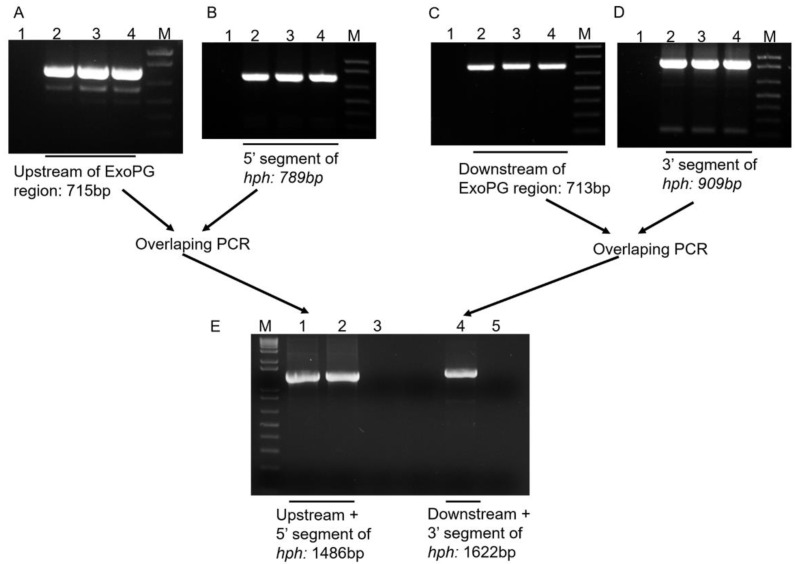
Construction of DNA fragment for knocking-out *ExoPG* gene in *V. dahliae* by overlapping PCR. (**A**) Amplification of upstream DNA region of *ExoPG* ORF; (**B**) Amplification of 5′ segment of hph gene from pSK846 vector; (**C**) Amplification of downstream DNA region of *ExoPG* ORF; (**D**) Amplification of posterior-half of hph gene from pSK846 vector; Lane 1 represents the negative control for PCR; Lane 2 to 4 represent the PCR products; Lane M represents the DNA marker (1Kb Plus DNA Ladder, Invitrogen, Waltham, MA, USA); (**E**) Connecting the PCR fragment by overlapping PCR; Lane 1 and 2 represent the upstream DNA connected with 5′ segment of hph by overlapping PCR; Lane 3 represents the negative control for PCR; Lane 4 represents the downstream DNA connected with 3′ segment of hph by overlapping PCR; Lane 5 represents the negative control for PCR; Lane M represent the DNA marker (1 Kb Plus DNA Ladder, Invitrogen, Waltham, MA, USA).

**Figure 4 pathogens-10-00642-f004:**
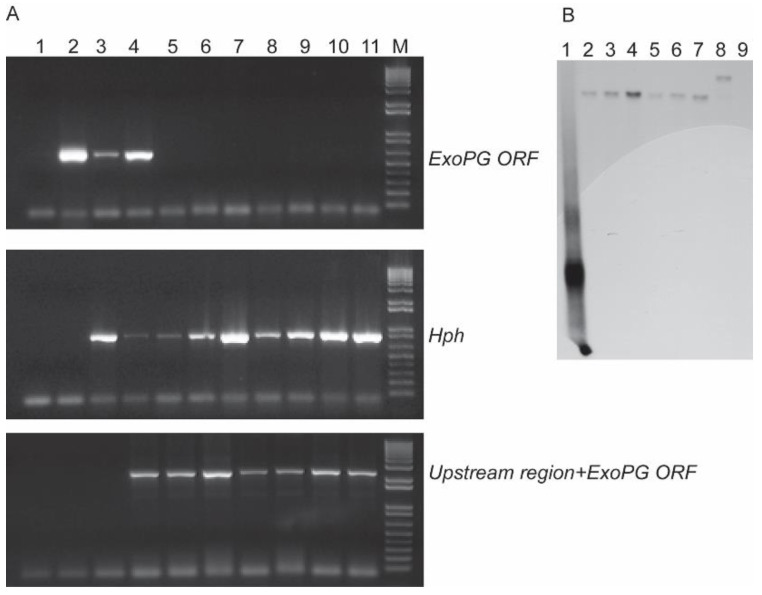
Identification of *△exopg* knock-out mutants. (**A**) PCR analysis of *△exopg* knocking-out transformants; Lane 1 represents the negative control for PCR; Lane 2 represents the genomic DNA of wild type Vd1396-9; Lane 3 to 11 represent transformants; Lane M represents the DNA marker (1 Kb Plus DNA Ladder, Invitrogen, Waltham, MA, USA); (**B**) Southern blot analysis of positive transformants of *△exopg* knocking-out transformants; Lane 1 represents the probe amplified from *hph* gene; Lane 2 to 7 represent single hygromycin gene replacement *△exopg* mutants; Lane 8 represents ectopic control; Lane 9 represents the wild type Vd1396-9.

**Figure 5 pathogens-10-00642-f005:**
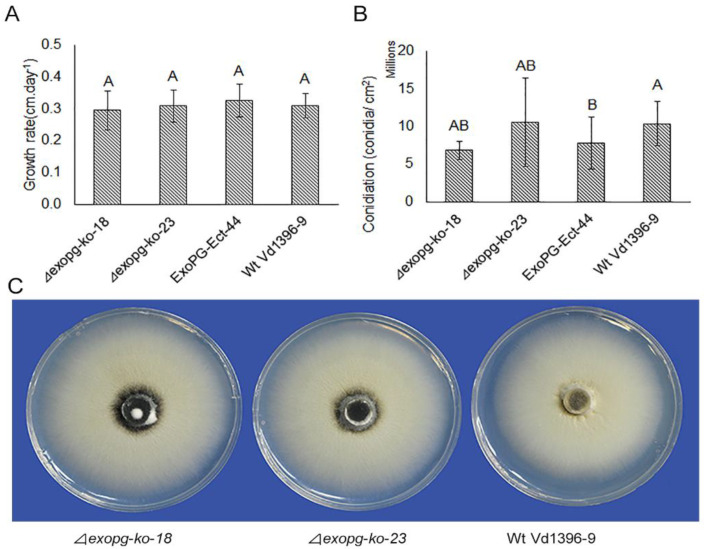
The phenotype analysis of *△exopg* mutants. Mycelial growth rate (**A**); conidiation (**B**); and colony phenotype (**C**) of wildtype (Vd1396-9) and mutant *(**△exopg-ko-18*, *△exopg-ko-23* and ExoPG-Ect-44) strains of *V. dahliae*. The bar graphs depict mean values (n = 8 for growth rate experiment, and n = 5 for conidiation experiment) ± standard error. Error bars refer to standard error. For each parameter, mean values marked by the different letters are significantly different according to multiple comparison (Fisher’s LSD) test (*p* < 0.05).

**Figure 6 pathogens-10-00642-f006:**
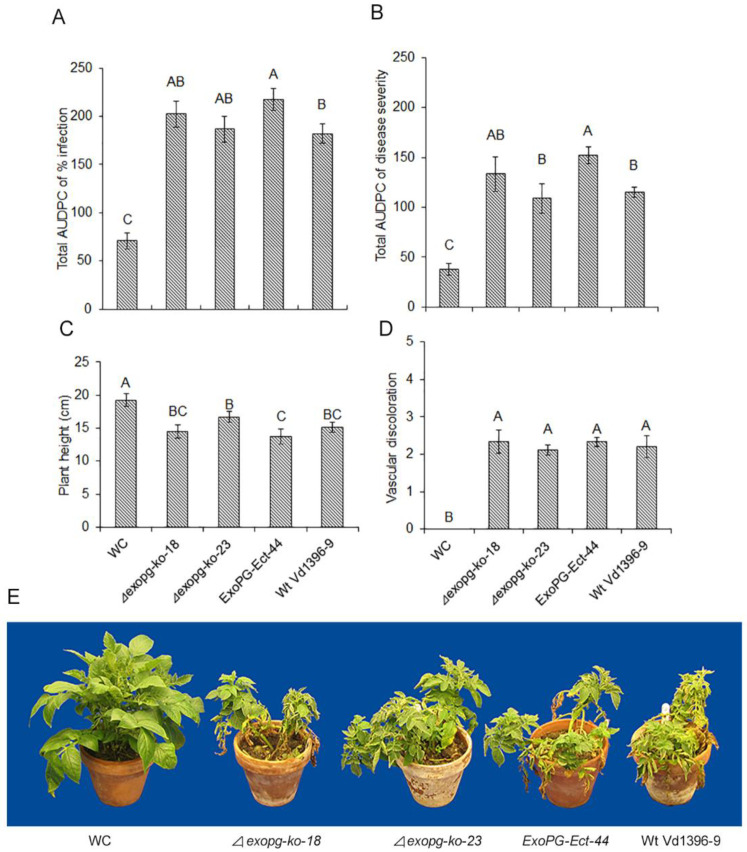
Pathogenicity test of *△exopg* mutants on potato cv. Kennebec. Total AUDPC of percentage of infection (**A**); total AUDPC of disease severity (**B**); height of potato plants (**C**); vascular discoloration rate (**D**); and samples of symptoms development on cv Kennebec at 5 weeks post-inoculation (**E**) with water (WC), wildtype (Vd1396-9) and mutant *(**△exopg-ko-18,*
*△exopg-ko-23*, *ExoPG-Ect-44*) strains of *V. dahliae*. Error bars refer to standard error. The bars represent mean values (n = 6) ± standard error. For each parameter, values with different letters represent significant differences between mean values according to multiple comparison (Fisher’s LSD) test (*p* < 0.05).

**Figure 7 pathogens-10-00642-f007:**
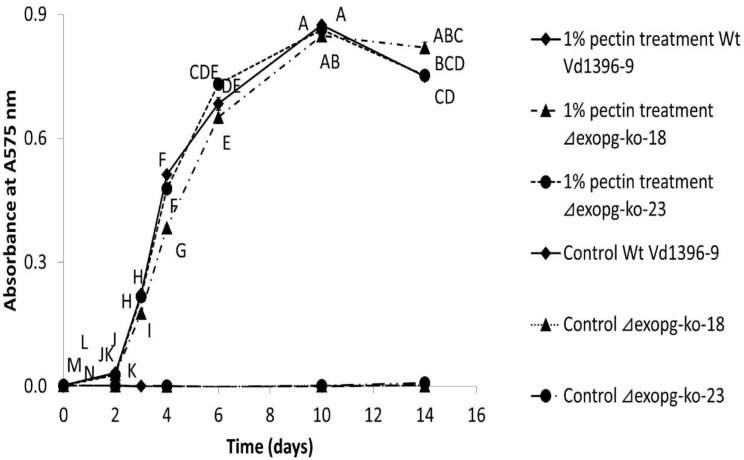
Total exopolygalacturonases activity of *△exopg* mutants (*△exopg-ko-18*, *△exopg-ko-23*) and wildtype (Vd1396-9). Graph points represent mean values (n = 3, with three technical replicates for each biological replicate) ± standard error. For each strain, means labelled by different letters are significantly different between treatments according to multiple comparison (Fisher’s LSD) test (*p* < 0.05). Error bars refer to standard error.

**Figure 8 pathogens-10-00642-f008:**
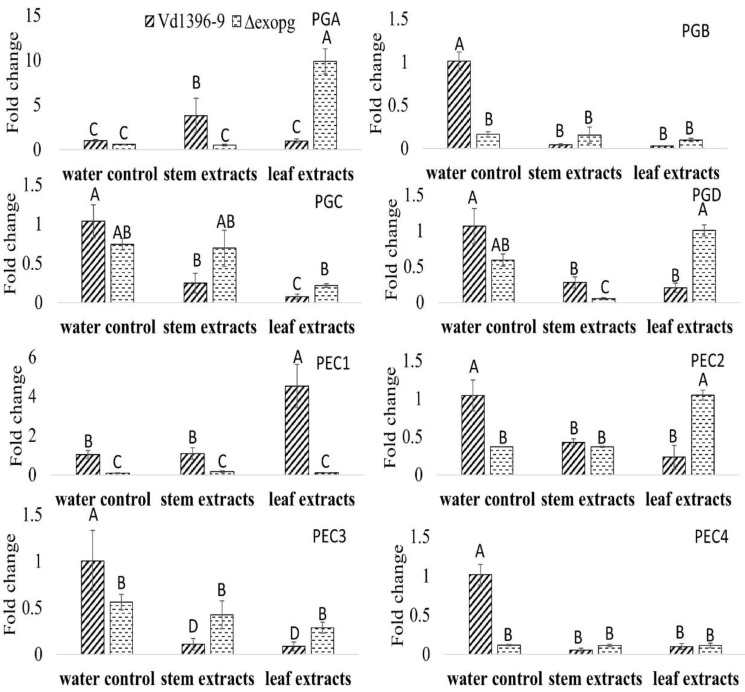
Expression of other polygalacturonase/pectinase genes in wildtype (Vd1396-9) and mutant *(**△exopg*) strains of *V. dahliae* in response to water, potato leaf and stem extracts. PGA: polygalacturonase A; PGB: polygalacturonase B; PGC: polygalacturonase C; PGD: polygalacturonase D; PEC 1: pectinase 1; PEC2: pectinase 2; PEC3: pectinase 3; and PEC4: pectinase 4. All QRT-PCR data were normalized with *V. dahliae* Histone H3. The bar values represent mean values (n = 3) ± standard error. For each gene, values marked by the same letters are not significantly different according to multiple comparison (Fisher’s LSD) test (*p* < 0.05).

**Figure 9 pathogens-10-00642-f009:**
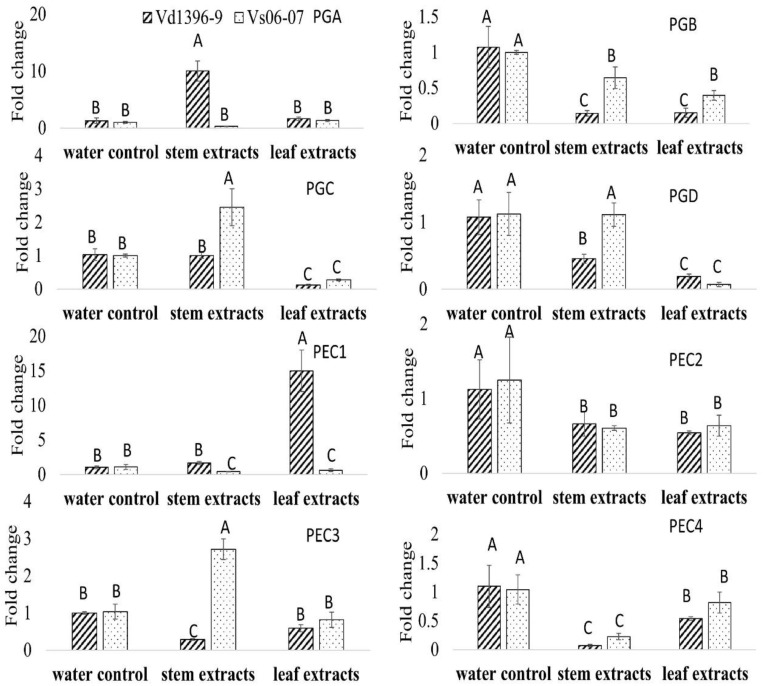
Expression of other polygalacturonase/pectinase genes in wildtype (Vd1396-9) and weakly aggressive isolate (vs-06-07) strains of *V. dahliae* in response to water, potato leaf and stem extracts: PGA: polygalacturonase A; PGB: polygalacturonase B; PGC: polygalacturonase C; PGD: polygalacturonase D; PEC 1: pectinase 1; PEC2: pectinase 2; PEC3: pectinase 3; and PEC4: pectinase 4. All QRT-PCR data were normalized with *V. dahliae* Histone H3. The bar values represent mean values (n = 3) ± standard error. For each gene, values marked by the same letters are not significantly different according to multiple comparison (Fisher’s LSD) test (*p* < 0.05).

**Table 1 pathogens-10-00642-t001:** The procedure of overlapping PCR.

Overlapping PCR	
PCR Mix 25 μL (Final Concentration)	Procedure
1x HF buffer	(1). Prepare the PCR mix
0.25 mM DNTPs	(2). 98 °C for 30 s
2 mM MgCl_2_	(3). 98 °C for 10 s
Phusion Taq 1 unit	(4). 49 °C for 90 s
	(5). 72 °C for 120 s
50 µg of PCR product of upstream/ or downstream region of the selected gene	Repeat step 3 to 5 for 8 cycles
	(6). 4 °C
	(7). Add primers (FS-AF/HY-R or YG-F/FS-BR) to a final concentration 0.2 uM
	(8). 98 °C for 30 s
	(9). 98 °C for 10 s
50 µg of PCR product of 5′/ or 3′ segment of *Hph* gene	(10). 58 °C for 90 s
	(11). 72 °C for 120 s
	Repeat step 8 to 10 for 30 cycles
	(12). 72 °C for 10 min
	(13). 4 °C

**Table 2 pathogens-10-00642-t002:** The primers in current study.

**Primer Name**	**5′ to 3′**	**Annealing** **Temperature (°C)**	**Accession Number**
ExoPG-QRT-F	GCCTTCCTCAACGACATCC	57.4	VDAG_03463
ExoPG-QRT-R	GAACTCACGCCACCAACG	57.4	VDAG_03463
FS-AF	GCCGTGTCAGTCAGAGGG	56.1	VDAG_03463
FS-AR	cctccactagctccagccaagcccaaaGGAAGTGTCACGAAACGC	56.1	VDAG_03463
FS-BF	agcactcgtccgagggcaaaggaatagACAAGACACGCCCAGGAC	57.8	VDAG_03463
FS-BR	CCAAAGTCGATTGAATGAAAT	57.8	VDAG_03463
ExoPG-ORF-F	GCACGGAGTACCAAAGG	54	VDAG_03463
ExoPG-ORF-R	GCAGCCAAGTCAGTAACAA	54	VDAG_03463
ExoPG-UA-F	TCACCTCACTATTATCCACCTC	55.1	VDAG_03463
Hph-TR [26]	GCTCCATACAAGCCAACC	55.1	Hph gene
Hph-F	TTTGGGCTTGGCTGGAGCTAGTGGA	55	Hph gene
Hph-R [27]	CTATTCCTTTGCCCTCGGACGAGT	55	Hph gene
YG-F [28]	GATGTAGGAGGGCGTGGATATGTCCT	55	Hph gene
HY-R [28]	GTATTGACCGATTCCTTGCGGTCCGAA	55	Hph gene

Note: *Hph* gene: hygromycin resistant gene.

**Table 3 pathogens-10-00642-t003:** Primers for other polygalacturonase/pectinase genes in *V. dahliae*.

**Primer Name**	**5′ to 3′**	**Annealing Temperature (°C)**	**Accession Number**
PGA-QRT-F	TGAGGATATCACGATGAAG	56.3	VDAG_07608
PGA-QRT-R	CCGCTGGGAACATTAATGG	55.4	VDAG_07608
PGB-QRT-F	AACAAGTCCGTCGAGGAG	55.9	VDAG_02879
PGB-QRT-R	ATGCCGTCGCAGATGATGA	58.2	VDAG_02879
PGC-QRT-F	AGTTCTTCGACAGTTCATCGG	56.7	VDAG_08097
PGC-QRT-R	GGACGCGGAAGGCATAATC	58.6	VDAG_08097
PGD-QRT-F	CTACGGCTTCATCAACAAC	52.8	VDAG_05992
PGD-QRT-R	AAAGAGAACGTTGGAGATGT	52.5	VDAG_05992
PEC-1-QRT-F	TCACTCAGTCTCGTCCTCCT	58.8	VDAG_09366
PEC1-QRT-R	GGATCCCTCCTCGAACAAAA	56.4	VDAG_09366
PEC2-QRT-F	CGGTGATGATTGCGTTTCG	56.3	VDAG_08098
PEC2-QRT-R	TGGGTTGGAAAGGCGTGTA	58.5	VDAG_08098
PEC3-QRT-F	CACCTTCAAGTACCAGTTCC	54.9	VDAG_01781
PEC3-QRT-R	TCGGTGACGAAGAGGATGG	58.1	VDAG_01781
PEC4-QRT-F	GCGTTCCTCGCAATATTCAA	54.7	VDAG_00768
PEC4-QRT-R	GGCAGATAGACAGCACCAC	57.2	VDAG_00768
His3-F [18]	ATGGCTCGCACTAAGCAA	54.8	VDAG_10035
His3-R [18]	TGAAGTCCTGGGCAATCT	52.7	VDAG_10035

## Data Availability

Accession numbers are listed in the body of the text.

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
