# Peer review of "Role of Exopolygalacturonase-Related Genes in Potato-Verticillium dahliae Interaction"

_pathogens, 2021, doi:10.3390/pathogens10060642_

Round 1
Reviewer 1 Report
Dear Authors,
My general impression of this study and its presentation in the form of a manuscript is positive, but some adjustments are necessary.
Arrange the manuscript following the journal's instruction. Discussion reports data from “materials and methods” and should be improved.
Title: In this form title is not appropriate to the manuscript. I suggest "Role of exopolygalacturonase-related genes in Potato-Verticillium dahliae interaction"
Line 51: delete ", both in citrus ".
Lines 55-57: use "virulence and indicates that pectin-degrading enzyme may play different functions and are involved in various processes [15]" Instead of " virulence ... pathogens [15]."
Lines 62-63: delete "Most ... ]."
Results
This section reports data from “materials and methods” (see lines 92-93; 172-175; Figure 3; 185-190; ...).
Tables are cited in the text but are not present.
I suggest the following results subsections:
2.1. ExoPG expression against potato extracts
2.2. Mutants production
2.3. Mutants characterization
2.3.a. Pathogenicity
2.3.b. Growth rate and conidiation
2.3.c. Exopolygalacturonases activity
2.4. Identification and expression of Polygalacturonase/pectinase-related genes
Figure 2: Are lane 1-5 the same in all the five gels?
I suggest the following legend: "Mycelial growth rate (A), conidiation (B) and colony phenotype (C) of wildtype (Vd1396-9) and mutant (△exopg-ko-18, △exopg-ko-23 and ExoPG-Ect-44) strains of V. dahliae. The bar graphs depict mean values (n = 8 for growth rate, and n=5 for conidiation) ± standard error. For each parameter, values with the same letters are not significantly different according to LSD test (p < 0.05)."
Figure 6: See figure 1 comments. Insert a space between "%" and "infection".
I suggest a legend like the following or similar: Total AUDPC of percentage of infection (A), total AUDPC of disease severity (B), height of potato plants (C), vascular discoloration rate (D) and samples of symptoms development on cv Kennebec at 5 weeks post-inoculation with water (WC), wildtype (Vd1396-9) and mutant (△exopg-ko-18, △exopg-ko-23, ExoPG-Ect-44) strains of V. dahliae. The bar graphs depict mean values (n = 6) ± standard error. For each parameter, values with the same letters are not significantly different according to LSD test (p < 0.05).
Figure 7: Reduce the space between thesis and increase letters size in legend. Insert axis titles and units.
I suggest the following legend: Exopolygalacturonases activity of wildtype (Vd1396-9) and mutant (△exopg-ko-18, △exopg-ko-23 and ExoPG-Ect-44) strains of V. dahliae. Graph points depict mean values (n = 3, with 3 technical replicates for each biological replicate) ± standard error. For each strain, values with the same letters are not significantly different according to LSD test (p < 0.05).
In alternative I suggest the following legend with symbols and delete legend in the graph: Exopolygalacturonases activity of V. dahliae strains Vd1396-9 (t), △exopg-ko-18 (p) and △exopg-ko-23 () grown in CDB medium alone (—) or amended with 1% Pectin (- - -). Graph points depict mean values (n = 3, with 3 technical replicates for each biological replicate) ± standard error. For each strain, values with the same letters are not significantly different according to LSD test (p < 0.05).
Lines 190-205: Explain the acronyms PGA, PGB, PGC, PGD, PEC1, PEC2, PEC3, PEC4.
Figure 8: See figure 1 comments. Magnify axis lines.
I suggest the following legend: Expression of other polygalacturonase/pectinase genes in wildtype (Vd1396-9) and mutant (△exopg) strains of V. dahliae in response to water, potato leaf and stem extracts: PGA (A), PGB (B), PGC (C), PGD (D), PEC1 (E), PEC2 (F), PEC3 (G), PEC4 (H). Water was used as control. All qRT-PCR data were normalized with V. dahliae Histone H3. The bar graphs depict mean values (n = 3) ± ???. For each gene, values with the same letters are not significantly different according to LSD test (p < 0.05)."
Materials and Methods
I suggest the following Materials and Methods subsections:
4.1. Fungal strains and plant material
4.2. ExoPG expression against potato extracts
4.3. Mutants production
4.3.a. Protoplast preparation and transformation
4.3.b. Vector-free split-marker recombination
4.4. Mutants characterization
4.4.a. Pathogenicity
4.4.b. Growth rate and conidiation
4.4.c. Exopolygalacturonases activity
4.5. Identification and expression of Polygalacturonase/pectinase-related genes
4.6. Statistical Analysis
Write "V. dahliae" in Italics (see lines 300, 305, 308, 311, 324, 326, ...)
Line 302: use "10 days" instead of "1.5 weeks"
Line 330: insert "filtration on" between "by" and "miracloth"
Line 330: put 4 as subscript (NaPO4)
Line 344: put 7 as apex (107)
Line 348: put 2 as subscript (CaCl2)
Line 348: use ")" instead of "))"
Is Figure 1 correct in Materials and Methods section?
Line 400: How vascular discoloration ratings were detected?
Line 449: What does it means "in this project"?
Line 456: What does it means "5. Conclusions"?
Line 463: What does it means "Please add:"?
Arrange the bibliography in the text and in reference section following the journal's instruction.
Best regards
Reviewer 2 Report
This paper tends to reveal the role of exopolygalacturonase-related genes in the pathogenicity of Verticillium dahliae to potato. In the present study, the authors have used a novel method ‘vector free spilt-marker recombination’ to construct △exopg mutants successfully for the study of gene function. However, there are several weaknesses in the experimental design, which should be addressed and improved to render the study publishable in this journal.
In general, the experiments for characterization of mutant, aggressiveness in planta, and gene expression studies have been conducted only once in this study. The English writing should be refined in this manuscript, especially in the method part. A lot of formatting, typing, and citing errors were found in the manuscript. Tables mentioned in the manuscript were not shown either in the main text or in supplemental documents.
As shown in H. Alkher's master dissertation, reference of this study, Vs06-07 isolate recovered from sunflowers was not always less aggressive than Vd1396-6 to susceptible cultivar Kennebec. Therefore, the result of disease assessment or fungal amount in the detached Kennebec potato leaves should be shown to support your conclusion in this study. Moreover, the host specificity of this isolate may have an effect on the interaction, instead of Vs 06-07 why not use other weakly aggressive isolates from potato, such as Vs04-09?
As you mentioned in the results and discussion, the expression of other polygalacturonase/pectinase genes were higher in △exopg than wild type. However, it would be also great to know, how these genes respond in highly and weakly aggressive isolates to potato extracts, in order to investigate whether such compensation could be found in weakly aggressive isolates as well.
Further weaknesses are shortly summarized as below and marked in detail on the PDF file.
- Formatting:
- Several gene abbreviations (ExoPG, Hph) and Latin names (V. dahliae) were not in italic in the text and legend of the figures.
- The abbreviation form of ExoPG gene should be identical throughout the manuscript, either ExoPG or Exo-PG.
- P for P value should be in capital letter and italic.
- The number in the chemical formula should be downscaled, e.g. MgSO4
- no space need between number and °C and same for %, space before other units, e.g. mM, ml, etc. Space between pH and number.
- Concentration should be up-scaled as 106.
- Some of the numberings of titles were in bold and some were not.
- Line 90-97:
- The order of the figures would be more logical if you put figure 3 in front of figure 2.
- Figure 7: What is the unit of the y-axis?
- Line 307: How were these isolates cultured, in which medium, under which condition, and pre-cultured for how long?
- Figure 8: stem extracts instead of stem estracts.
- Line 315: Which concentration of the spore suspension was used?
- Line 320: Table 1 is not shown in the manuscript. Please add it to the method part.
- Line 321: What was the melting temperature and how long was elongation time?
- Line 330-358: check the space units and the chemical formula
- Line 367-385: Figures 2 and 3 should be cited instead of Figure 1.
- Line 376: Which temperature was used for combination and overlapping PCR?
- Line 393: How long was the inoculation time?
- Line 440: Table 3 was not shown in the present manuscript. The sequence of all primers should be listed in a table or in a supplemental document.
- Line 444: Based on the MIQE guidelines to achieve a robust result for publication, at least two reference genes should be used for qRT-PCR.

Reviewer 3 Report
Dear authors,
this paper reported interesting results o the role of exopolygalacturonase-related genes in the pathogenicity of an ascomycete pathogen.
After reading the whole paper, minor corrections are detected: in figure 5 B, the legend AB for the second bar need to be place under the error bar; the grid pattern in most of graph for the condition Vd 1396-9 need to be replaced by another pattern, that will be better to observed; Figure 5C, the colony phenotype : specify the date of the colony? the same for the 3 culture?; for the figure 6E, the description of the plant morphology is strongly limited in results section, add more comments: do you measurement the effect on the growth by measuring the weight of the plant? the size of the leaves, other growth parameters?
Round 2
Reviewer 2 Report
The quality of the manuscript has been significantly improved. However, there are several points that need to be addressed and improved to render the study publishable in this journal.
The main problems were about the results and discussion. The data of gene expression shown in the new version were completely different from the former version, instead of up-regulated, most of the genes found down-regulated. Therefore, it is hard to address the conclusion made by the authors. Detailed information showed as in PDF.
